# Orbital-dependent electron correlation in double-layer nickelate La$_3$Ni$_2$O$_7$

Jiangang Yang[1,2,8], Hualei Sun[3,8], Xunwu Hu[4,8], Yuyang Xie[1,2], Taimin Miao[1,2], Hailan Luo [1,2], Hao Chen[1,2], Bo Liang[1,2], Wenpei Zhu[1,2], Gexing Qu[1,2], Cui-Qun Chen [4], Mengwu Huo[4], Yaobo Huang[5], Shenjin Zhang[6], Fengfeng Zhang[6], Feng Yang[6], Zhimin Wang[6], Qinjun Peng[6], Hanqing Mao[1,2,7], Guodong Liu[1,2,7], Zuyan Xu[6], Tian Qian [1], Dao-Xin Yao [4] ✉, Meng Wang [4] ✉, Lin Zhao [1,2,7] ✉ & X. J. Zhou [1,2,7] ✉

The latest discovery of high temperature superconductivity near 80 K in La$_3$Ni$_2$O$_7$ under high pressure has attracted much attention. Many proposals are put forth to understand the origin of superconductivity. The determination of electronic structures is a prerequisite to establish theories to understand superconductivity in nickelates but is still lacking. Here we report our direct measurement of the electronic structures of La$_3$Ni$_2$O$_7$ by high-resolution angle-resolved photoemission spectroscopy. The Fermi surface and band structures of La$_3$Ni$_2$O$_7$ are observed and compared with the band structure calculations. Strong electron correlations are revealed which are orbital- and momentum-dependent. A flat band is formed from the Ni-3d$_{z^2}$ orbitals around the zone corner which is ~ 50 meV below the Fermi level and exhibits the strongest electron correlation. In many theoretical proposals, this band is expected to play the dominant role in generating superconductivity in La$_3$Ni$_2$O$_7$. Our observations provide key experimental information to understand the electronic structure and origin of high temperature superconductivity in La$_3$Ni$_2$O$_7$.

The superconductivity in cuprates is realized by doping the Mott insulators and strong electron correlation is believed to be essential to produce high temperature superconductivity[1–4]. Great efforts have been made to search for high temperature superconductivity in nickelates which have similar structural and electronic characteristics as cuprates[5–9]. The latest discovery of superconductivity near 80 K in double-layer nickelate La$_3$Ni$_2$O$_7$ under pressure has attracted enormous attention[10–12]. Many theoretical proposals are put forward to

understand the origin of superconductivity in La$_3$Ni$_2$O$_7$[13–38]. However, direct determination of the electronic structures is still lacking which is a prerequisite for establishing theories to understand superconductivity in La$_3$Ni$_2$O$_7$.

In this paper, we measured the electronic structure of La$_3$Ni$_2$O$_7$ at ambient pressure by using angle-resolved photoemission spectroscopy (ARPES). We observed its Fermi surface and band structures and compared them with the band structure calculations. A Ni-3d$_{z^2}$

[1]Beijing National Laboratory for Condensed Matter Physics, Institute of Physics, Chinese Academy of Sciences, Beijing 100190, China. [2]School of Physical Sciences, University of Chinese Academy of Sciences, Beijing 100049, China. [3]School of Science, Sun Yat-Sen University, Shenzhen, Guangdong 518107, China. [4]Guangdong Provincial Key Laboratory of Magnetoelectric Physics and Devices, School of Physics, Sun Yat-Sen University, Guangzhou 510275, China. [5]Shanghai Synchrotron Radiation Facility, Shanghai Advanced Research Institute, Chinese Academy of Sciences, Shanghai 201204, China. [6]Technical Institute of Physics and Chemistry, Chinese Academy of Sciences, Beijing 100190, China. [7]Songshan Lake Materials Laboratory, Dongguan, Guangdong 523808, China. [8]These authors contributed equally: Jiangang Yang, Hualei Sun, Xunwu Hu. ✉e-mail: yaodaox@mail.sysu.edu.cn; wangmeng5@mail.sysu.edu.cn; LZhao@iphy.ac.cn; XJZhou@iphy.ac.cn

orbital-derived flat band is observed around the zone corner which is ~50 meV below the Fermi level. La$_3$Ni$_2$O$_7$ exhibits orbital- and momentum-dependent electron correlations and the Ni-3d$_{z^2}$ derived band shows much stronger electron correlation than the Ni-3d$_{x^2-y^2}$ derived bands.

## Results

### Calculated band structures of La$_3$Ni$_2$O$_7$

La$_3$Ni$_2$O$_7$ crystallizes in an orthorhombic phase (space group *Amam*) at ambient pressure, with a corner-connected NiO$_6$ octahedral layer separated by a La-O fluorite-type layer stacking along the c axis (Fig. 1a)[39,40]. Due to the out-of-plane tilting of the Ni-O octahedras, the original Ni-O plaquette (solid black frame in Fig. 1b) is reconstructed into a two-Ni unit cell (dashed black frame in Fig. 1b) which doubles the volume of the original unit cell. Correspondingly, in the reciprocal space, this structural reconstruction results in the shrinking of the first Brillouin zone to half of the original one. Figure 1c shows such a folded three-dimensional first Brillouin zone with high-symmetry points and high-symmetry momentum lines marked.

We carried out band structure calculations of La$_3$Ni$_2$O$_7$ based on the DFT+U method (see Methods). Figure 1d, e shows calculated band structures of La$_3$Ni$_2$O$_7$ without ($U$ = 0, Fig. 1d) and with ($U$ = 3.5 eV, Fig. 1e) considering the effective on-site Coulomb interaction $U$. The electronic states around the Fermi level come mainly from the Ni-3d$_{x^2-y^2}$ and Ni-3d$_{z^2}$ orbitals. The Ni-3d$_{z^2}$ orbital-derived bands form two branches. The upper branch represents the anti-bonding bands ($\gamma_{an}$) while the lower branch represents the bonding bands ($\gamma$) originated from the strong inter-layer coupling between the two Ni-3d$_{z^2}$ orbitals via the O-2p$_z$ orbitals of the intermediate apical oxygen. Such a coupling results in a large energy splitting between the two branches which gives rise to a separation (between $\gamma$ and $\gamma_{an}$ in Fig. 1d) of ~0.2 eV. The Ni-3d$_{z^2}$ orbitals form a flat band ($\gamma$ band in Fig. 1d) around $\Gamma$ and it slightly crosses the Fermi level. The Ni-3d$_{x^2-y^2}$ orbitals form two kinds of bands ($\alpha$ and $\beta$ in Fig. 1d, e) due to the in-plane coupling of the Ni-3d$_{x^2-y^2}$ orbitals via the intermediate O-2p$_{x/y}$ orbitals. Both the $\alpha$ and $\beta$ bands cross the Fermi level.

When the effective on-site Coulomb interaction is turned on ($U$ = 3.5 eV, Fig. 1e), it causes a small effect on the overall band structures of La$_3$Ni$_2$O$_7$ around the Fermi level when compared with the case of $U$ = 0 (Fig. 1d). The upper branch of the Ni-3d$_{z^2}$ derived bands, as well as the Ni-3d$_{x^2-y^2}$ derived $\alpha$ and $\beta$ bands, exhibits a weak change with or without considering U. The most notable effect is on the lower branch of the Ni-3d$_{z^2}$ derived bands. Upon turning on U, these bands increases in the band width and the overall energy position shifts downwards to the high binding energy. This results in an obvious change of the $\gamma$ band near $\Gamma$. It crosses the Fermi level for $U$ = 0 (Fig. 1d) but sinks ~50 meV below the Fermi level for $U$ = 3.5 eV (Fig. 1e).

### Fermi surface of La$_3$Ni$_2$O$_7$

Figure 2 shows the Fermi surface mapping of La$_3$Ni$_2$O$_7$ measured at 18 K by using both synchrotron (Fig. 2a) and laser (Fig. 2b) light sources. The typical band structures along high-symmetry directions are presented in Fig. 3. In Fig. 2a, b, the original first Brillouin zone is marked by the solid black line and the corresponding folded first Brillouin zone is marked by the dashed black line. Two main Fermi surface sheets are clearly observed, as quantitatively shown in Fig. 2c. Due to the lattice distortion (Fig. 1b), the original $\alpha$ and $\beta$ Fermi surface are folded to form $\alpha'$ and $\beta'$ Fermi surface, as observed in Fig. 2a and plotted in Fig. 2c. In addition, we also observed features (green dashed line in Fig. 2a) that cannot be attributed to the $\alpha$ and $\beta$ Fermi surface or their folded Fermi surface $\alpha'$ and $\beta'$. This feature is observed in synchrotron-based ARPES measurements (Fig. 2a) but not in the laser-based ARPES measurements with a small spot size of ~15 $\mu$m (Fig. 2b and Supplementary Fig. 1). The feature is not expected in the band structure calculations (Fig. 1d, e). We therefore tend to believe that it may originate from other impurity phase in the measured sample. Since the extra feature follows the same Brillouin zone as La$_3$Ni$_2$O$_7$, it is possible that it comes from intergrowth phase.

The measured Fermi surface of La$_3$Ni$_2$O$_7$ consists of an electron-like $\alpha$ sheet and a hole-like $\beta$ sheet (Fig. 2c). This Fermi surface topology is similar to that of La$_4$Ni$_3$O$_{10}$[41] regarding the Ni-3d$_{x^2-y^2}$ orbital derived $\alpha$ and $\beta$ Fermi surface. One obvious difference is that the $\gamma$

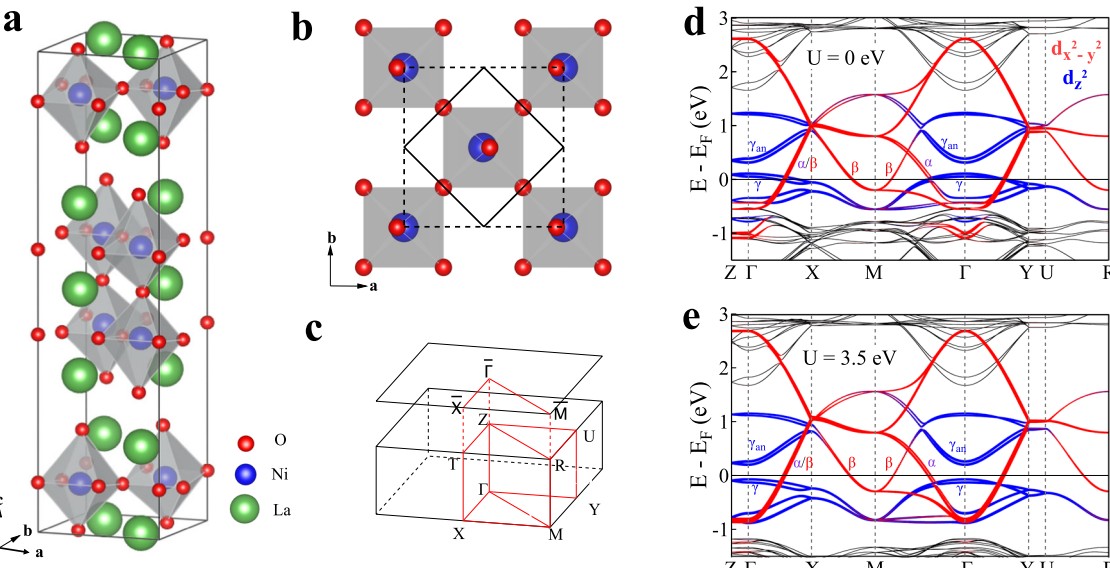

**Fig. 1 | Calculated band structures of La$_3$Ni$_2$O$_7$. a** Schematic pristine crystal structure of La$_3$Ni$_2$O$_7$. **b** Top view of the crystal structure with a two-dimensional lattice of O and Ni atoms. Due to the out-of-plane tilted Ni-O octahedra, the apical oxygen atom is not right on the top of Ni atom. The solid black line frame represents the original unit cell without considering the tilted Ni-O octahedra and the dashed black line frame represents the real structural unit cell by considering the tilting of the Ni-O octahedra. **c** Three-dimensional Brillouin zone with high-symmetry points and high-symmetry momentum lines marked which are obtained based on the real structural unit cell (dashed frame in **b**). **d** and (**e**) Calculated band structures of La$_3$Ni$_2$O$_7$ without considering $U$ (**d**) and with $U$ = 3.5 eV (**e**). Red color represents 3d$_{x^2-y^2}$ orbital of Ni while blue color represents 3d$_{z^2}$ orbital of Ni.

band crosses the Fermi level in $La_4Ni_3O_{10}$ forming the $\gamma$ Fermi surface[41] while it stays below the Fermi level in $La_3Ni_2O_7$ without forming a Fermi surface.

The measured Fermi surface of $La_3Ni_2O_7$ is not compatible with the band structure calculations with $U = 0$ (Fig. 1d). Without considering U, the $\gamma$ band crosses the Fermi level and forms a $\gamma$ Fermi

surface around $\Gamma$. This is different from the measured Fermi surface where the $\gamma$ band lies below the Fermi level (Fig. 3b) without the formation of $\gamma$ Fermi surface. To reconcile the discrepancy, the effective on-site Coulomb interaction U has to be added in the band structure calculations. With $U = 3.5$ eV, the $\gamma$ band around $\Gamma$ shifts downwards to -50 meV below the Fermi level which is consistent with the measured position of the $\gamma$ band (Fig. 3b). Also the calculated Fermi surface (Fig. 2d) agrees well with the measured one (Fig. 2c). The addition of sizable U in the band structure calculations indicates that there is a strong effect of Coulomb interaction in $La_3Ni_2O_7$.

From the area of the measured Fermi surface (Fig. 2c), the doping level of the $\alpha$ sheet is estimated to be -0.20 electron/Ni and the doping level of the $\beta$ sheet corresponds to -1.25 hole/Ni. This results in the overall doping of -0.95 electron/Ni. From the calculated Fermi surface in Fig. 2d, the doping levels of the $\alpha$ and $\beta$ Fermi surfaces are 0.25 electron/Ni and 1.25 hole/Ni, respectively. This gives an overall doping level of 1.0 electron/Ni. The measured doping levels show a good agreement with the calculated ones. They indicate that in $La_3Ni_2O_7$ the Ni-$3d_{x^2-y^2}$ orbital derived bands are doped with 1 electron/Ni.

## Measured band structures of $La_3Ni_2O_7$

Figure 3 shows band structures of $La_3Ni_2O_7$ measured along several high-symmetry directions. The observed bands are marked by the guidelines. We took momentum cuts in both the first and second Brillouin zones because, due to the photoemission matrix element effects, the observed bands may show different spectral intensity even though the momentum cuts are equivalent in the momentum space. The $\alpha$ band is observed along the $\bar{\Gamma}$-$\bar{M}$ momentum cut (Cut4, Fig. 3e). The $\beta$ band is observed around $\bar{M}$ along the $\bar{M}$-$\bar{\Gamma}$-$\bar{M}$ momentum cut (Cut1, Fig. 3b). Along the $\bar{\Gamma}$-$\bar{X}$-$\bar{\Gamma}'$ momentum cut (Cut3, Fig. 3d), the $\alpha$ and $\beta$ bands are nearly degenerates and clearly observed. The $\gamma$ band are observed around $\bar{\Gamma}$ $(\pi,\pi)$ along the $\bar{M}$-$\bar{\Gamma}$-$\bar{M}$ momentum cut (Cut1, Fig. 3b) and the $\bar{\Gamma}$-$\bar{X}$-$\bar{\Gamma}'$ momentum cut (Cut2, Fig. 3c). The corresponding photoemission spectra (energy dispersive curves, EDCs) are shown in Fig. 3f. The $\gamma$ band is nearly flat around $\bar{\Gamma}$ $(\pi,\pi)$ over a large momentum region. It lies -50 meV below the Fermi level. We found that the energy position of the flat band slightly changes between 30 ~ 50 meV in different samples we measured. This is likely due to the sample inhomegeneity, particularly the oxygen content variation in

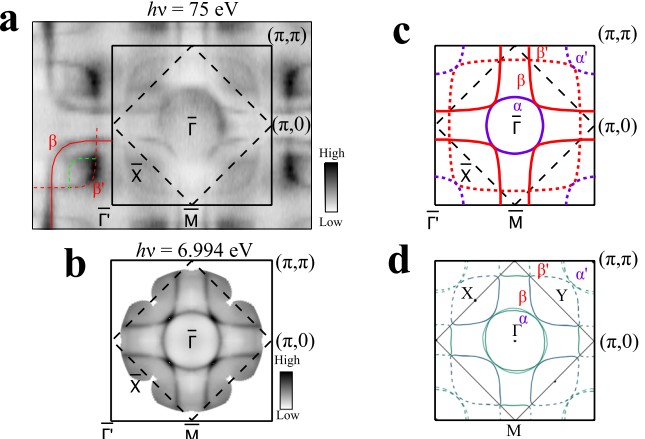

**Fig. 2 | Measured Fermi surface of $La_3Ni_2O_7$ and its comparison with the band structure calculations. a** Fermi surface mapping measured at 18 K by using synchrotron-based ARPES with a photon energy of 75 eV. It is obtained by integrating the spectral intensity within 20 meV with respect to the Fermi level. **b** Fermi surface mapping measured at 18 K by using laser-based ARPES with a photon energy of 6.994 eV. It is obtained by integrating the spectral intensity within 10 meV with respect to the Fermi level. The Fermi surface mappings in (**a**) and (**b**) are obtained by symmetrization assuming four-fold symmetry. Two main Fermi surface sheets ($\alpha$ and $\beta$) are clearly observed. The solid line frame represents the first Brillouin zone from the original unit cell (solid line frame in Fig. 1**b**) while the dashed line frame represents the first Brillouin zone from the real unit cell (dashed line frame in Fig. 1**b**). **c** Measured Fermi surface of $La_3Ni_2O_7$ obtained from (**a**) and (**b**). It consists of two main Fermi surface sheets, $\alpha$ and $\beta$, and their folded Fermi surface, $\alpha'$ and $\beta'$. **d** Calculated Fermi surface with $U = 3.5$ eV obtained from the first-principles calculations (Fig. 1**e**). The calculated Fermi surface shows an excellent agreement with the measured Fermi surface in (**c**).

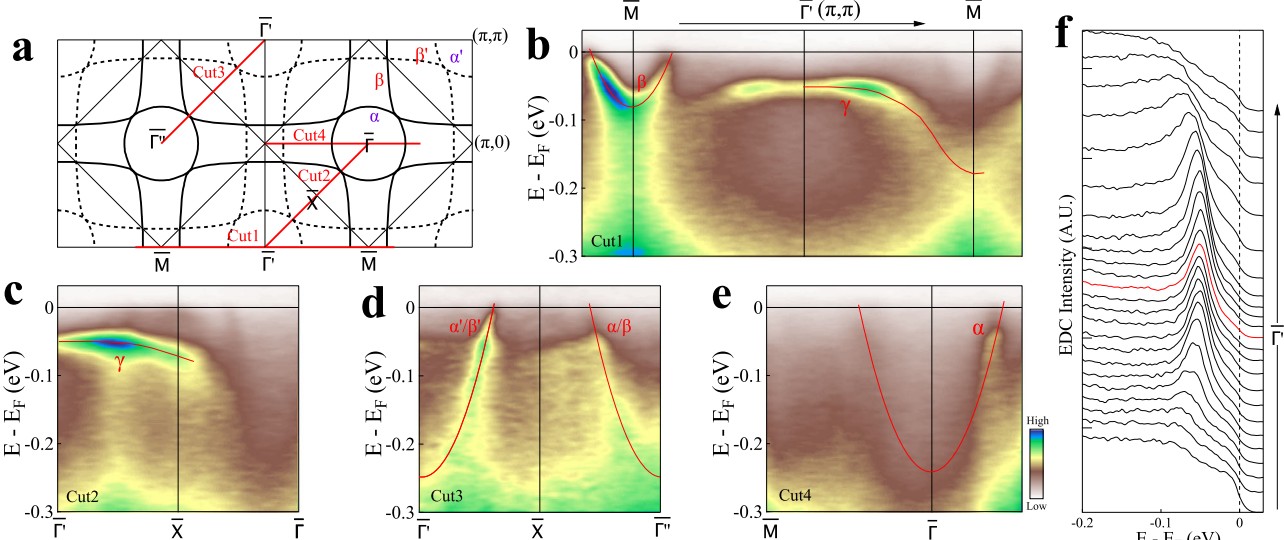

**Fig. 3 | Band structures of $La_3Ni_2O_7$ measured at 18 K along high-symmetry directions from synchrotron-based ARPES with a photon energy of 75 eV. a** Schematic Fermi surface of $La_3Ni_2O_7$ with the momentum cuts marked. **b**−**e** Band structures measured along momentum cuts Cut1, Cut2, Cut3 and Cut4,

respectively. The location of the momentum cuts is shown by red lines in (**a**). The observed bands are labeled by their corresponding Fermi surface and shown by guidelines. **f** EDC stack of the flat band $\gamma$ from (**b**). The momentum region is marked by the arrowed line on top of (**b**).

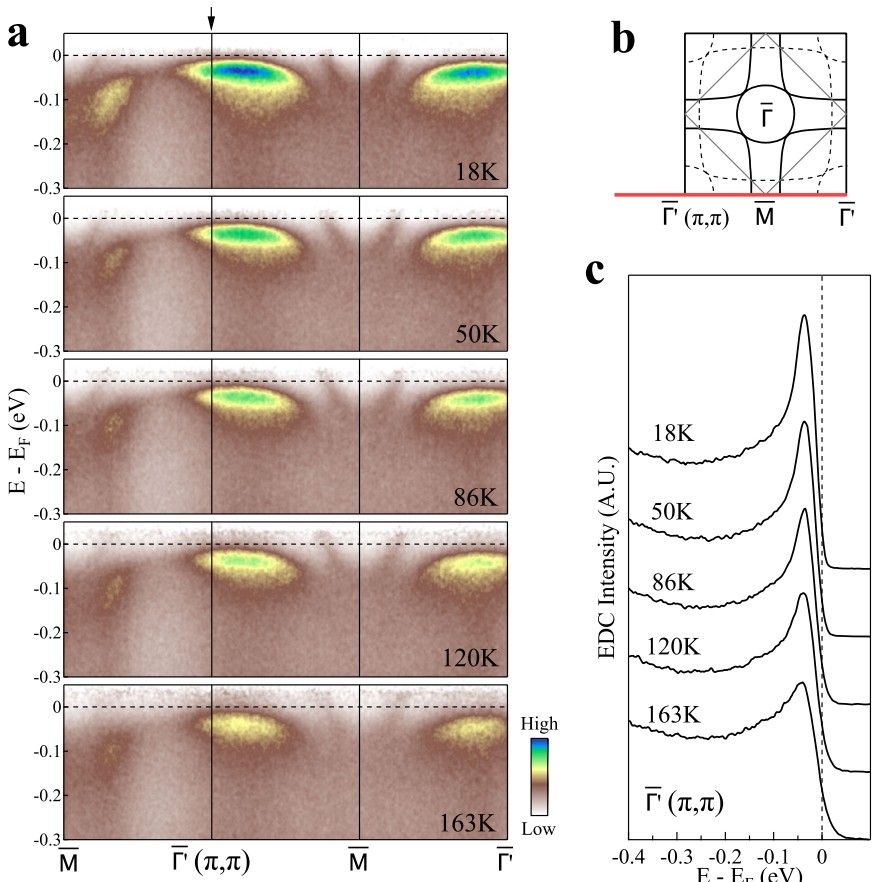

**Fig. 4 | Temperature dependence of the γ flat band in La₃Ni₂O₇. a** Band structures measured along the $\bar{\Gamma}'$-$\bar{M}$-$\bar{\Gamma}'$ direction at different temperatures using 85 eV photon energy. The location of the momentum cut is shown by the red line in (**b**). **b** Schematic Fermi surface of La₃Ni₂O₇ with the momentum cut marked. **c** EDCs at the $\bar{\Gamma}'(\pi,\pi)$ point at different temperature from **a**. The corresponding momentum postion is marked by the arrow in **a**.

La₃Ni₂O₇₋δ[42]. We note that there is a strong spectral weight buildup below the observed bands. This is particularly clear for the β band around $\bar{M}$ in Fig. 3b. They can also be observed for the α/β bands in Fig. 3d. These observations are similiar to the waterfall band structures observed in cuprate superconductors which can be attributed to the strong electron correlations in the measured materials[43–48].

To investigate the orbital characters of different bands in La₃Ni₂O₇, we carried out polarization-dependent ARPES measurements as shown in Supplementary Fig. 1. Based on the analysis of the photoemission matrix element effects as described in Supplementary Note 1, the results are consistent with the orbital assignment that the α/β bands are dominated by the Ni-3$d_{x^2-y^2}$ orbital while the γ band is dominated by the Ni-3$d_{z^2}$ orbital.

### Temperature dependence of the γ flat band in La₃Ni₂O₇

In order to check on the nature of the γ flat band in La₃Ni₂O₇, we carried out temperature-dependent measurements of the band as shown in Fig. 4. These data are measured with 85 eV photon energy. The temperature-dependent band structures along the $\bar{\Gamma}'$-$\bar{M}$-$\bar{\Gamma}'$ direction are shown in Fig. 4a and the extracted EDCs at $\bar{\Gamma}'(\pi,\pi)$ are plotted in Fig. 4c. The flat band changes little with temperature and stays at the similar position (~40 meV) below the Fermi level over the temperature range of 18 ~ 163 K. It was found that La₃Ni₂O₇ exhibits a resistivity anomaly at ~110 K and a magnetic susceptibility anomaly at ~153 K[40]. Our results indicate that the flat band in La₃Ni₂O₇ does not change in its energy position both across the resistivity anomaly temperature and across the susceptibility anomaly temperature. This is different from the case of La₄Ni₃O₁₀. In La₄Ni₃O₁₀, the flat band near (π,π) crosses the Fermi level above the resistivity anomaly temperature (~140 K). But it opens a gap of ~20 meV below the resistivity anomaly temperature and stays ~20 meV below the Fermi level at low temperature[41]. The different behaviors of the flat band near (π,π) make La₃Ni₂O₇ distinct from La₄Ni₃O₁₀.

### Band renormalizations in La₃Ni₂O₇

Figure 5 shows a quantitative comparison between the measured bands and the calculated ones with $U=0$ (Fig. 5a–c) and $U=3.5$ eV (Fig. 5d, e). Here we present the results of all the α, β and γ bands along two high-symmetry directions: one is along the axes while the other is along the diagonal as shown in the inset of Fig. 5d. It is clear that the calculated bands are obviously wider than the measured ones. By renormalizing the calculated bands with a factor to match the measured dispersions, we obtain the mass renormalization factors for different bands measured along different momentum directions as shown in Fig. 5f. The mass renormalization is found to be strongly orbital-dependent. The Ni-3$d_{x^2-y^2}$ derived α and β bands exhibit relatively weak band renormalization (~2) and they are nearly isotropic in the momentum space. On the other hand, the Ni-3$d_{z^2}$ derived γ band shows a strong band renormalization (5 ~ 8) which is also momentum-dependent. These indicate the Ni-3$d_{z^2}$ derived γ band shows much stronger electron correlation than the Ni-3$d_{x^2-y^2}$ derived α and β bands in La₃Ni₂O₇. It is usually considered that, if an energy band is fully occupied, there is no correlation effects and hence no band renormalization. It is therefore interesting that in La₃Ni₂O₇, although the Ni-3$d_{z^2}$-derived flat band lies below the Fermi level, it exhibits strong band renormalization effect. Our density functional theory calculations reveal that, in this bilayer phase, the two nearest intra-layer Ni cations exhibit significant interlayer coupling through two Ni-3$d_{z^2}$ orbitals via

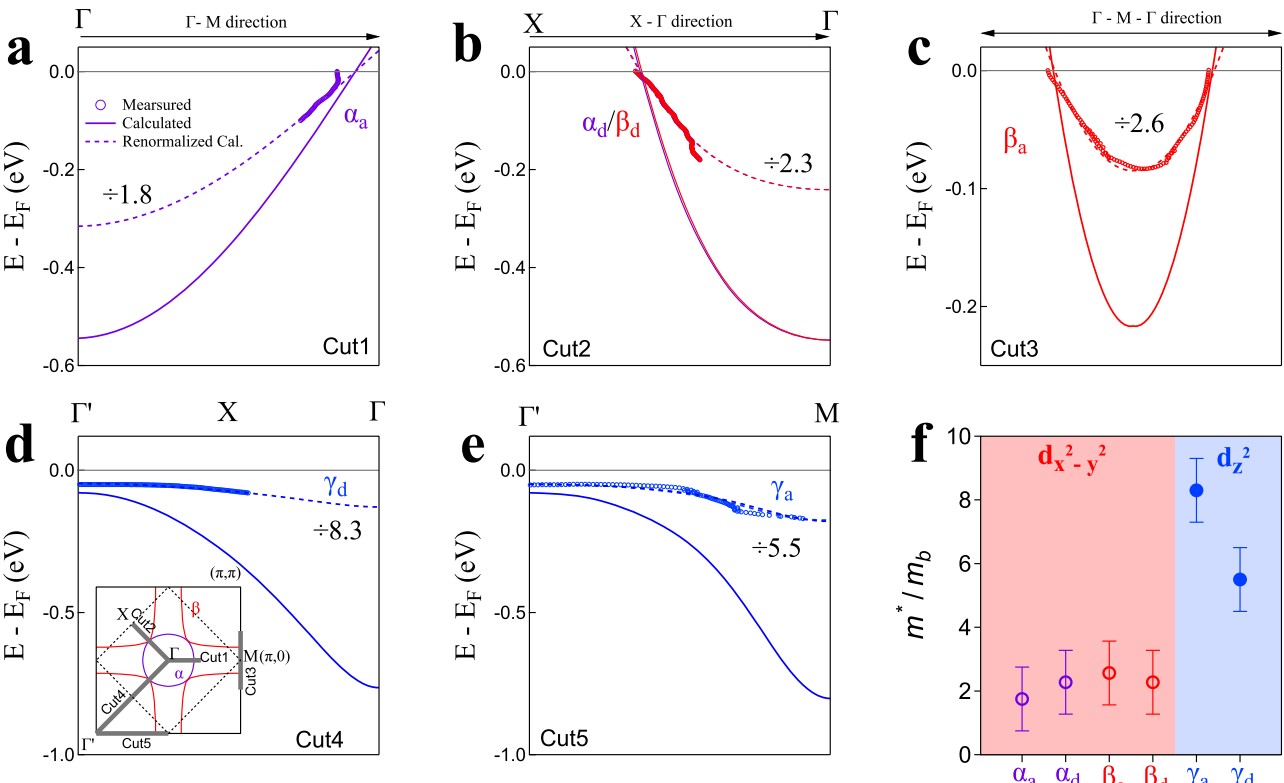

**Fig. 5 | Orbital- and momentum-dependent band renormalization in La₃Ni₂O₇.**
**a**–**e** Measured band dispersions (empty circles) and the corresponding calculated bands (solid lines) from Fig. 1d,e along the momentum cuts Cut1-Cut5, respectively. The location of the momentum cuts is marked by solid gray lines in the inset of (**d**). To match the observed dispersions, the dashed curves are the renormalized bands obtained from the calculated bands scaled by the corresponding mass enhancement values as shown in each panel. **f** Measured mass enhancements of the $\alpha$, $\beta$ and $\gamma$ bands. For each band, it is further divided into two directions, along the axes ($\alpha_a$, $\beta_a$ and $\gamma_a$) and along the diagonal direction ($\alpha_d$, $\beta_d$ and $\gamma_d$). The error bars reflect the uncertainty in determining the band renormalization factors.

the apical oxygen. This coupling arises from the quantum confinement of the NiO₂ bilayer within the structure, resulting in an energy splitting of Ni cations that makes the Ni-$3d_{z^2}$ bonding bands lower in energy and fully occupied. However, for each Ni cation, only one electron is occupied in the Ni-$3d_{z^2}$ orbital. Consequently, the Ni-$3d_{z^2}$ orbital is half-filled, leading to the effects of band renormalization in La₃Ni₂O₇.

## Discussion

The high temperture superconductivity in La₃Ni₂O₇ is realized by applying high pressure[10]. It is found that there is a pressure-induced phase transition from the orthorhombic phase with a space group *Amam* to another orthorhombic phase with a space group *Fmmm*[10]. This is accompanied by a change of the bond angle of Ni-O-Ni from 168.0° at ambient pressure to 180° at high pressure along the c axis and a dramatic reduction of the inter-atomic distance between the Ni and apical oxygen from 2.297 Å at ambient pressure to 2.122 Å at high pressure[10]. The key question is which orbitals play the dominant role in producing superconductivity in La₃Ni₂O₇ and whether the strong electron correlations are involved.

Our measured electronic structures of La₃Ni₂O₇ at ambient pressure are consistent with the calculated results including sizable effective on-site Coulomb interaction. We found that there are strong electron correlations in La₃Ni₂O₇ which are orbital- and momentum-dependent. The Ni-$3d_{z^2}$ derived $\gamma$ band shows much stronger electron correlation than the Ni-$3d_{x^2-y^2}$ derived $\alpha$ and $\beta$ bands. It supports the picture that the Ni-$3d_{z^2}$ orbital is more localized. The ferromagnetic Hund's rule coupling J may also contribute to the strong band renormalization because of its competition with the hybridization between Ni-$3d_{z^2}$ orbitals and Ni-$3d_{x^2-y^2}$ orbitals. Since Ni-$3d_{z^2}$ orbitals are half-filled in La₃Ni₂O₇, like the Cu-$3d_{x^2-y^2}$ orbitals in the parent cuprate

compounds, the observed strong band renormalization of the $\gamma$ band may be related to the Mott physics. On the other hand, DMFT calculations[30] indicate that increasing the onsite Coulomb repulsion U alone does not significantly enhance band renormalization. It does not produce an obvious orbital-selectivity either. Only when the Hund's rule coupling J is considered can the orbital-selective strong band renormalization be realized[30]. Our observation of strong orbital-selective band renormalizations indicates that the Hund physics is at play in La₃Ni₂O₇. These features show that La₃Ni₂O₇ is a unique compound with orbital-selective Mott and Hund physics.

Since it is difficult to measure the electronic structure of La₃Ni₂O₇ under high pressure by ARPES due to technical limitation, we carried out DFT+U calculation at 29.5 GPa and compared it with that at ambient pressure (as shown in Supplementary Fig. 2). Band structure calculations indicate that the $\gamma$ bands derived from the Ni-$3d_{z^2}$ orbital undergo a pronounced change in the energy position under pressure; it is below the Fermi level at ambient pressure but crosses the Fermi level under high pressure. On the other hand, its band width only slightly increases under high pressure primarily due to the enhanced inter-layer interaction. The band renormalization effect is expected to exhibit a small change with pressure and the electron correlation effect remains strong under high pressure. This is consistent with many theoretical proposals that the Ni-$3d_{z^2}$ orbital-derived flat band is expected to play the dominant role in generating superconductivity in La₃Ni₂O₇[10,16,18,19,22,24].

For a new unconventional superconductor like La₃Ni₂O₇, the determination of its electronic structures is a prerequisite to establish theories to understand superconductivity. Understanding the electronic structure of La₃Ni₂O₇ at ambient pressure is essential for understanding its electronic structure under high pressure. Since

$La_3Ni_2O_7$ is a strongly correlated system, it is uncertain how well its electronic structures can be represented by the band structure calculations. Our first ARPES measurements are necessary and significant to provide direct electronic structure information to check on the reliability of band structure calculations and establish theories to understand superconductivity in $La_3Ni_2O_7$.

## Methods

### Growth of single crystals

The single crystals of double-layer nickelate $La_3Ni_2O_7$ were grown by using a high-pressure floating zone method[10,40]. Typical sample size is ~1 mm. It is noted that $La_3Ni_2O_7$ single crystal may become weakly insulating at ambient pressure due to the oxygen deficiency[49,50]. Therefore, our samples are annealed under high oxygen pressure of 100 atmospheres at 500°C before ARPES experiments. After the annealing, the magnetic measurement shows the consistent result with that in ref. 40.

### ARPES measurements

Synchrotron-based ARPES measurements were performed at the beamline BL09U and BL03U of the Shanghai Synchrotron Radiation Facility (SSRF) with a hemispherical electron energy analyzer DA30L (Scienta-Omicron). The energy resolution was set at 10 ~ 15 meV. High-resolution ARPES measurements were also performed using a lab-based ARPES system equipped with the 6.994 eV vacuum-ultra-violet (VUV) laser and a hemispherical electron energy analyzer DA30L (Scienta-Omicron)[51,52]. The energy resolution was set at 1 meV and the angular resolution was 0.3 degree. The momentum coverage is increased by applying bias on the sample during the ARPES measurements[53] (Supplementary Fig. 1). All the samples were cleaved in situ at a low temperature of 18 K and measured in ultrahigh vacuum with a base pressure better than $5 \times 10^{-11}$ mbar. The Fermi level is referenced by measuring on clean polycrystalline gold that is electrically connected to the sample.

### Band structure calculations

The first-principles calculations are performed based on the density functional theory as implemented in the Vienna ab initio simulation package (VASP)[54,55]. The generalized gradient approximation (GGA) of Perdew-Burke-Ernzerhof (PBE)[56] form is used for exchange-correlation functional. The projector augmented-wave (PAW) potential[57] with a 600 eV plane-wave cutoff energy is employed. A $\Gamma$-centered $19 \times 19 \times 5$ $k$-points mesh with Monkhorst-Pack scheme is used for self-consistant and Fermi-surface calculations. The lattice parameters are fixed to the experimentally refined lattice constants[58], and the atomic positions are fully optimized until the forces on each atom are $<10^{-3}$ eV/Å, and the energy convergence criterion is set to be $10^{-6}$ eV.

To address the effect of strong Coulomb interaction, we employed DFT+U method as described in ref. 59. To determine the U parameter, we tested the $U$-values with 3, 3.5 and 4 eV. The position of the $\gamma$ band closely matches the experimental measurement (~50 meV below the Fermi level) when $U$ is set to 3.5 eV. Consequently, an effective Hubbard $U = 3.5$ eV is taken for the 3d electrons of Ni cations in this work.

## Data availability

All data are processed by using Igor Pro 8.02 software. All data needed to evaluate the conclusions in the paper are available within the article. All raw data generated during the current study are available from the corresponding author upon request.

## Code availability

The codes used for the DFT calculations in this study are available from the corresponding authors upon request.

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

## Acknowledgements

We thank Prof. Guangming Zhang, Prof. Yifeng Yang and Zhihui Luo for fruitful discussions, and Zhengtai Liu for beamline support. This work is supported by the National Key Research and Development Program of China (Grant No. 2021YFA1401800, 2018YFA0704200, 2022YFA1604200, 2022YFA1403800, 2022YFA1402802 and 2018YFA0306001), the National Natural Science Foundation of China (Grant No. 12488201, 11974404, 12074411, 12174454, 92165204 and U22A6005), the Strategic Priority Research Program (B) of the Chinese Academy of Sciences (Grant No. XDB25000000 and XDB33000000), Innovation Program for Quantum Science and Technology (Grant No. 2021ZD0301800), the Youth Innovation Promotion Association of CAS (Grant No. Y2021006), Synergetic Extreme Condition User Facility (SECUF), the Informatization Plan of Chinese Academy of Sciences (CAS-WX2021SF-0102), the Guangdong Basic and Applied Basic Research Foundation (Grant No. 2021B1515120015), the Guangzhou Basic and Applied Basic Research Funds (Grant No. 2024A04J6417), the Guang-dong Provincial Key Laboratory of Magnetoelectric Physics and Devices (Grant No. 2022B1212010008), Shenzhen International Quantum Acad-emy and National Supercomputer Center in Guangzhou.

## Author contributions

J.G.Y., H.L.S. and X.W.H. contribute equally to this work. X.J.Z., L.Z., J.G.Y. and M.W. proposed and designed the research. H.L.S., M.W.H. and M.W. contributed to single crystal growth and characterizations. X.W.H., C.Q.C and D.X.Y. contributed to the DFT band calculations. J.G.Y. and

Y.Y.X. carried out the ARPES experiment. T.M.M., H.L.L., H.C., B.L., W.P.Z., S.J.Z, F.F.Z., F.Y., Z.M.W., Q.J.P., H.Q.M., G.D.L., L.Z., Z.Y.X. and X.J.Z. contributed to the development and maintenance of Laser-ARPES system. Y.B.H., Q.G.X. and Q.T. contributed to the experimental assistance in SSRF. J.G.Y., L.Z., D.X.Y. and X.J.Z. analyzed the data. J.G.Y., L.Z. and X.J.Z. wrote the paper. All authors participated in the discussion and comment on the paper.

## Competing interests

The authors declare no competing interests.
