## [Peer Review File · Nature Communications]

REVIEWER COMMENTS

Reviewer #1 (Remarks to the Author):

Yang, Sun, and Hu et al report the best electronic structure measurements in bilayer nickelate La₃Ni₂O₇ to date, obtained on bulk single crystals. Their main observation is an extra shallow and flat band near (π,π), mainly consisting of d_{z^2} content. The authors derive the significance of their work mainly from the large renormalization of the d_{z^2} band, and its putative connection to the superconducting high pressure phase. The report is sound and straightforward, but I am unsure - based on the above understanding - how much this adds to the understanding of the superconducting phase, especially in the context of prior related literature, and many unexplained band features in the current observation.

First, the observation of a flat, shallow gamma band near (π,π) is not unique to La₃Ni₂O₇. For example, in La₄Ni₃O₁₀ systems, a flat shallow band near (π,π) is also present (ref[41]). Given the very similar structure motif and method, one naturally expect such a band in the 327 system. Moreover, key to the perceived significance of this work, the authors discussed the possibility of gamma band crossing E_F in the pressurized 327 phase, which leads to superconductivity. However, this condition is satisfied in 4310 system, but no superconductivity is observed. In this regard, I feel the relevance of the work is undermined at least in its current presentation.

Second, it is puzzling to see the lack of distinction between alpha and beta bands along G-X/Y directions, especially given the very high energy momentum resolution provided in this work. Looking at Ref[41], which has supposedly worse energy-momentum resolution, alpha-beta bands are both quite distinguishable. In addition, looking at Fig. 1e and Fig. 2d, it is not clear why there is a near-generate alpha-beta band crossing in the FS map, but no such a crossing at E_F in the calculated energy-momentum cut along G-X/Y directions. This seems like an apparent inconsistency within the calculations. Perhaps I am massively confused by the presentation in some way?

Third, the authors seem to blame impurity phase for a FS sheet (smaller than beta/beta') that crosses the ($\pi,0$)-(0, π) folded zone boundary. However, I find this explanation unsatisfactory - if it is an impurity phase, why would it obey the folded zone boundary of the primary La₃27 phase, and why is it almost concentric to the beta pockets? If the significance of this work is based on a comprehensive mapping of the electronic structure in La₃27 bulk system, then this aspect is crucial to be further addressed.

Then, there is yet another mysterious hole band that seems to just touch the $X^{\bar{}}$ point in Fig.3. What is that? Can these cleaved surfaces be so polar/inhomogenous that surface order and additional states form? Then one needs slab calculations and surface electron diffraction to clarify this issue.

In addition, the mass renormalization discussion is incomplete. Akin to such analysis in iron-based superconductors (e.g. npj Quantum Materials 2, 57), the correlation-induced renormalization should only flatten bands (at most in addition to a global chemical potential shift due to doping effect). However, it should not cause nonuniform changes in the Fermi crossing momenta from band to band. This requirement is clearly violated in Fig.4.

Last but not least, assuming the mission of the work is to correctly identify the bands and their respective orbital content (before discussing renormalizations due to correlation), the discussion in Fig.3 is insufficient to achieve this goal. Currently, Fig.3 shows a seemingly random choice of 4 cuts across different BZs. It could be a good opportunity to discuss the orbital character when combined with careful matrix element analysis. However, without such discussions/simulations, it is unclear what types of orbital content distribution can make cuts 2/3 and 1/4 look so different.

Reviewer #2 (Remarks to the Author):

In the manuscript, the authors performed an ARPES measurement on $\text{La}_3\text{Ni}_2\text{O}_7$ which had been found to show superconductivity near 80K under pressure. Compared with the first-principles calculations, the author found that the Ni-3d_{z^2} orbital forms a flat band-like structure and is the most strongly correlated, which may be a key to understanding the high temperature superconductivity of this compound.

Superconductivity near 80K has a great impact on the field of condensed matter physics, and clarifying the electronic structure of $\text{La}_3\text{Ni}_2\text{O}_7$ is an essential and urgent task. Since ARPES is one of the most powerful tools for observing the electronic structure in a compound, the manuscript may contain valuable information for a wide range of audiences. However, the relation between the observed spectrum and the high temperature superconductivity is not clearly analyzed. In my view, the authors need to address the following points.

1. All experiments in this manuscript were performed under ambient pressure, while the superconductivity was observed under high pressure. The authors just briefly address the effect of pressure in the discussion section, citing their previous publication. Suppose the authors want to stress the importance of the orbital-dependent correlation effect on superconductivity. In that case, they should expand their discussions further, such as how mass renormalization will change (or unchange) under pressure, etc.

2. Computational details of how the authors treated on-site Coulomb interaction ($U=3.5\text{eV}$) on top of DFT calculation must be addressed.

3. The authors stress the orbital-selective Mott and Hund physics in the discussion section. Regarding this, is there any indication of Mott/Hund physics from the current experimental result?

Reviewer #3 (Remarks to the Author):

This work reports an ARPES study on the recently discovered superconductor $\text{La}_3\text{Ni}_2\text{O}_7$, which shows a T_c as high as 80 K under high pressure. They successfully observed energy bands originating from the Ni $3d_{x^2-y^2}$ and Ni $3d_{z^2}$ orbitals and concluded that the d_{z^2} bands show stronger correlation than the $d_{x^2-y^2}$ bands. As this is the first ARPES study on the compound, which has attracted strong interest recently, and the experiment has been properly carried out, I recommend its publication in Nature Communication if the following points are reconsidered in a revised manuscript:

1. The authors state that they have included the effect of electron correlation in their LDA+U calculation. However, by the LDA+U method, one can include the effect of Coulomb interaction only on the mean-field level, and no electron correlation (i.e., fluctuations around the mean-field) can be included. Otherwise, the authors could not have used the LDA+U band structure as a starting point for estimating the band renormalization. Also, "the LDA/DFT+U method" has to be mentioned in the manuscript.

2. It has been known that, if the energy band is fully occupied, there is no correlation effects and hence no band renormalization. Therefore, because the d_{z^2} band is gapped and the lower band is fully occupied, estimating the band renormalization using the same method as that for a metallic band may need some remarks.

3. Any possible impurity phase that gave rise to the Fermi surface indicated in green? Since the ARPES data show that the lattice constant of the impurity phase is identical to $\text{La}_3\text{Ni}_2\text{O}_7$, an intergrowth of another Ruddlesden-Popper layers?

A minor comment:

4. It is true that the d_{z^2} character is dominant in the corner of the extended Brillouin zone scheme, but according to in the present folded zone scheme and the present experimental data, the zone "center" seems more realistic terminology?

Response to Reviewer's Comments

Response to Reviewer #1's comments

Yang, Sun, and Hu et al report the best electronic structure measurements in bilayer nickelate La₃Ni₂O₇ to date, obtained on bulk single crystals. Their main observation is an extra shallow and flat band near (π,π), mainly consisting of d_{z^2} content. The authors derive the significance of their work mainly from the large renormalization of the d_{z^2} band, and its putative connection to the superconducting high pressure phase. The report is sound and straightforward, but I am unsure - based on the above understanding - how much this adds to the understanding of the superconducting phase, especially in the context of prior related literature, and many unexplained band features in the current observation.

We thank the reviewer for the careful reviewing of our paper and providing constructive comments and suggestions to improve our paper. We thank the reviewer for pointing out that this is the best electronic structure measurement in La₃Ni₂O₇ and the report is sound and straightforward. For a new unconventional superconductor like La₃Ni₂O₇, the determination of its electronic structures is a prerequisite to establish theories to understand superconductivity. Understanding the electronic structure of La₃Ni₂O₇ at ambient pressure is important for understanding its electronic structure under high pressure. Since La₃Ni₂O₇ is a strongly correlated system, it is uncertain how well its electronic structures can be represented by the band structure calculations. Our first ARPES measurements are necessary and significant to provide direct electronic structure information to check on the reliability of band structure calculations and establish theories to understand superconductivity in La₃Ni₂O₇.

Following the reviewer's suggestion, in the revised manuscript, we added more discussion on the importance of electronic structure determination of La₃Ni₂O₇ at ambient pressure in understanding its superconductivity under high pressure: “For a new unconventional superconductor like La₃Ni₂O₇, the determination of its electronic structures is a prerequisite to establish theories to understand superconductivity. Understanding the electronic structure of La₃Ni₂O₇ at ambient pressure is essential for understanding its electronic structure under high pressure. Since La₃Ni₂O₇ is a strongly correlated system, it is uncertain how well its electronic structures can be represented by the band structure calculations. Our first ARPES measurements are necessary and significant to provide direct electronic structure information to check on the reliability of band structure calculations and establish theories to understand superconductivity in La₃Ni₂O₇.”

First, the observation of a flat, shallow gamma band near (π,π) is not unique to La₃Ni₂O₇. For example, in La₄Ni₃O₁₀ systems, a flat shallow band near (π,π) is also present (ref[41]). Given the very similar structure motif and method, one naturally expect such a band in the 327 system. Moreover, key to the perceived significance of

this work, the authors discussed the possibility of gamma band crossing EF in the pressurized 327 phase, which leads to superconductivity. However, this condition is satisfied in 4310 system, but no superconductivity is observed. In this regard, I feel the relevance of the work is undermined at least in its current presentation.

It is true that one may expect a flat band in $\text{La}_3\text{Ni}_2\text{O}_7$ near (π, π) based on the result of $\text{La}_4\text{Ni}_3\text{O}_{10}$. But without our ARPES measurements, no one can know for sure the energy position of the flat band and its band renormalization in $\text{La}_3\text{Ni}_2\text{O}_7$. Our results indicate that the flat band in $\text{La}_3\text{Ni}_2\text{O}_7$ is $\sim 50\text{meV}$ below the Fermi level at low temperature. In many theoretical proposals, this flat band is considered to play a dominant role in generating superconductivity in $\text{La}_3\text{Ni}_2\text{O}_7$ [10, 16, 18, 19, 22, 24]. Band structure calculations indicate that this flat band moves up and crosses the Fermi level under high pressure [10]. In $\text{La}_4\text{Ni}_3\text{O}_{10}$, the flat band near (π, π) crosses the Fermi level above the resistivity anomaly temperature ($\sim 140\text{K}$) [41]. But it opens a gap of $\sim 20\text{meV}$ and stays below the Fermi level at low temperature ($\sim 20\text{K}$). Therefore, there is no inconsistency in the observation of superconductivity in $\text{La}_3\text{Ni}_2\text{O}_7$ under high pressure and the absence of superconductivity in $\text{La}_4\text{Ni}_3\text{O}_{10}$ at ambient pressure in terms of the Fermi level crossing of the flat band.

Following the reviewer's comment, we carried out more ARPES experiments on the temperature-dependence of the γ flat band in $\text{La}_3\text{Ni}_2\text{O}_7$. It is added as a new figure (Figure 4) in the revised manuscript and replotted in Figure R1. We found that the flat band in $\text{La}_3\text{Ni}_2\text{O}_7$ stays below the Fermi level both at low temperature and above the resistivity anomaly temperature ($\sim 110\text{K}$) and the susceptibility anomaly temperature ($\sim 153\text{K}$). This new result of the flat band makes $\text{La}_3\text{Ni}_2\text{O}_7$ distinct from $\text{La}_4\text{Ni}_3\text{O}_{10}$.

Fig. R1. Temperature-dependent measurement of the γ flat band in $\text{La}_3\text{Ni}_2\text{O}_7$, replotted from Fig. 4. a Band structures measured along the $\bar{\Gamma}' - \bar{M} - \bar{\Gamma}'$ direction at different temperatures using 85eV photon energy. The location of the momentum cut is shown by the red line in **b**. **b** Schematic Fermi surface of $\text{La}_3\text{Ni}_2\text{O}_7$ with the

momentum cut marked. **c** EDCs at the $\bar{\Gamma}'$ (π,π) point at different temperature from **a**. The corresponding momentum position is marked by the arrow in **a**.

Second, it is puzzling to see the lack of distinction between alpha and beta bands along G-X/Y directions, especially given the very high energy momentum resolution provided in this work. Looking at Ref [41], which has supposedly worse energy-momentum resolution, alpha-beta bands are both quite distinguishable. In addition, looking at Fig. 1e and Fig. 2d, it is not clear why there is a near-degenerate alpha-beta band crossing in the FS map, but no such a crossing at EF in the calculated energy-momentum cut along G-X/Y directions. This seems like an apparent inconsistency within the calculations. Perhaps I am massively confused by the presentation in some way?

There is no inconsistency between our data in $\text{La}_3\text{Ni}_2\text{O}_7$ and the data in Ref [41] in $\text{La}_4\text{Ni}_3\text{O}_{10}$; the α and β bands are nearly degenerate along Γ -X/Y directions in both cases. In $\text{La}_3\text{Ni}_2\text{O}_7$, the measured α and β bands are nearly degenerate along Γ -X/Y directions as seen in Fig. 3d and replotted in Fig. R2a. This is also consistent with the band structure calculations where the α and β bands are nearly degenerate along Γ -X/Y directions as shown in Fig. 1e and replotted in Fig. R2b. In $\text{La}_4\text{Ni}_3\text{O}_{10}$, the measured α and β bands are nearly degenerate along the exact Γ -X/Y directions as seen in Supplementary Figure 1b in Ref [41] and replotted in Fig. R3b. These two bands become separated and distinguishable when the momentum cut moves away from the exact Γ -X/Y directions, as seen in Supplementary Figure 1d in Ref [41] and replotted in Fig. R3c. There is no measured data in Ref [41] showing that the α and β bands are distinguishable exactly along the Γ -X/Y directions.

Fig. R2. **a** Band structures of $\text{La}_3\text{Ni}_2\text{O}_7$ measured along Γ -X/Y directions. It is reproduced from Fig. 3d. **b** Calculated band structures of $\text{La}_3\text{Ni}_2\text{O}_7$ with $U=3.5$ eV. It is reproduced from Fig. 1e. The α and β bands are nearly degenerate as marked by the green arrow.

Fig. R3. **a** Schematic Fermi surface of $\text{La}_4\text{Ni}_3\text{O}_{10}$ and location of two momentum cuts marked by green line and blue line. It is replotted from Supplementary Fig. 1a of Ref [41]. **b** Band structure taken along the exact Γ -X/Y direction as marked by the green line in **a**. It is replotted from Supplementary Fig. 1b of Ref [41]. **c** Band structure taken along the momentum cut as marked by the blue line in **a**. It is replotted from Supplementary Fig. 1c of Ref [41].

There is no inconsistency within our calculations. In Fig. 2d, as replotted in Fig. R4a, there is a nearly degenerate α - β band crossing in the Fermi surface mapping along Γ -X/Y directions, as marked by the green arrow in Fig. R4a. This corresponds to the two nearly degenerate α/β bands along Γ -X/Y directions in Fig. 1e, as replotted in Fig. R4b and marked by the green arrow.

Fig. R4. **a** Calculated Fermi surface of $\text{La}_3\text{Ni}_2\text{O}_7$ with $U = 3.5 \text{ eV}$. It is replotted from Fig. 2d. The α and β Fermi surface nearly touch each other along Γ -X/Y directions as marked by the green arrow. **b** Calculated band structures of $\text{La}_3\text{Ni}_2\text{O}_7$ with $U = 3.5 \text{ eV}$. It is reproduced from Fig. 1e. The α and β bands are nearly degenerate as marked by the green arrow.

Third, the authors seem to blame impurity phase for a FS sheet (smaller than beta/beta') that crosses the $(\pi,0)$ - $(0,\pi)$ folded zone boundary. However, I find this explanation unsatisfactory - if it is an impurity phase, why would it obey the folded zone boundary of the primary La327 phase, and why is it almost concentric to the beta pockets? If the significance of this work is based on a comprehensive mapping of the electronic structure in La327 bulk system, then this aspect is crucial to be further addressed.

Following the reviewer's suggestion, we carried out more ARPES measurements. The results are added in the Supplementary Information as Supplementary Fig. 1 and replotted as Fig. R5. We carried out laser ARPES measurements on different places of the sample surface and also on different samples of La₃Ni₂O₇. Since our laser has a small spot size of $\sim 15\mu\text{m}$, it is possible to reduce the effect of impurity phase. As seen in Supplementary Fig. 1 and Fig. R5, we do not observe the extra feature seen in synchrotron ARPES measurement marked as dashed green line in Fig. 2a. These results further indicate that this extra feature is unlikely to be attributed to intrinsic electronic structures of La₃Ni₂O₇ but more likely to originate from impurity phase.

Fig. R5. High-resolution laser ARPES measurements of La₃Ni₂O₇. It is replotted from Supplementary Fig. 1 in the revised manuscript.

Then, there is yet another mysterious hole band that seems to just touch the $X^{\bar{}}$ point in Fig.3. What is that? Can these cleaved surfaces be so polar/inhomogenous that surface order and additional states form? Then one needs slab calculations and surface electron diffraction to clarify this issue.

The hole band that touches the \bar{X} point in Fig. 3 corresponds to the extra feature seen in the Fermi surface mapping in Fig. 2a. As shown in Fig. R6, the extra hole band in Fig. R6a corresponds to the extra feature marked by the dashed blue line in Fig. R6b. It has the same origin as the extra feature marked by the dashed green line in Fig. R6b. As we have discussed before, they have the same origin that are more likely from impurity phase.

Fig. R6. **a** Band structures of $\text{La}_3\text{Ni}_2\text{O}_7$ measured along $\bar{\Gamma} - \bar{X} - \bar{\Gamma}'$ (momentum Cut2) in Fig. 3. It is replotted from Fig. 3c. An extra hole band is observed near \bar{X} as marked by the blue arrow. **b** Fermi surface mapping of $\text{La}_3\text{Ni}_2\text{O}_7$. It is replotted from Fig. 2a. The extra feature (dashed blue line) corresponds to the extra hole band in **a**. It has the same origin as the extra feature marked by the dashed green line.

In addition, the mass renormalization discussion is incomplete. Akin to such analysis in iron-based superconductors (e.g. npj Quantum Materials 2, 57), the correlation-induced renormalization should only flatten bands (at most in addition to a global chemical potential shift due to doping effect). However, it should not cause nonuniform changes in the Fermi crossing momenta from band to band. This requirement is clearly violated in Fig.4.

We thank the reviewer for the comment. We agree with the reviewer that usually the correlation-induced renormalization should only flatten bands. This is true for the single band system. For the multi-band systems, the situation may become more complicated. As shown in Fig. 1d,e and replotted in Fig. R7, when the Coulomb interaction U is considered, the Fermi momenta of the α , β and γ bands change when compared with the case of $U=0$. Also the three bands show different energy shift from $U=0$ to $U=3.5$ eV cases.

Our density functional theory calculations reveal that, in $\text{La}_3\text{Ni}_2\text{O}_7$, the two nearest intra-layer Ni cations exhibit significant interlayer coupling through two $3d_{z^2}$ orbitals via the apical oxygen. This coupling arises from the quantum confinement of the NiO_2

bilayer within the structure, resulting in an energy splitting of Ni cations that makes the $3d_{z^2}$ bonding bands lower in energy and fully occupied.

Following the reviewer's comments, we think it is more reasonable to compare the measured γ band with the calculated one by considering $U=3.5$ eV. In this case, the calculated γ band lies below the Fermi level. In the revised manuscript, we replaced the calculated γ band ($U=0$) with the calculated one ($U=3.5$ eV), and re-estimated the renormalization factors. This change has little effect on the obtained renormalization values and does not affect the main conclusions of our work.

Fig. R7. **a** Calculated band structures of $\text{La}_3\text{Ni}_2\text{O}_7$ without considering U . **b** Calculated band structures of $\text{La}_3\text{Ni}_2\text{O}_7$ with $U=3.5$ eV. Red color represents $3dx^2-y^2$ orbital of Ni while blue color represents $3dz^2$ orbital of Ni. It is replotted from Fig. 1d,e.

Last but not least, assuming the mission of the work is to correctly identify the bands and their respective orbital content (before discussing renormalizations due to correlation), the discussion in Fig.3 is insufficient to achieve this goal. Currently, Fig.3 shows a seemingly random choice of 4 cuts across different BZs. It could be a good opportunity to discuss the orbital character when combined with careful matrix element analysis. However, without such discussions/simulations, it is unclear what types of orbital content distribution can make cuts 2/3 and 1/4 look so different

The four momentum cuts in Fig. 3 are chosen based on several requirements. (1). They are along high symmetry directions. (2). They include both diagonal direction and the axial direction. (3). All the three bands can be clearly observed.

We noticed that there is strong matrix element effects in the ARPES measurements of $\text{La}_3\text{Ni}_2\text{O}_7$. Although cut 2 and cut 3 are equivalent in the momentum space, they show quite different matrix element effects because cut 2 lies in the first BZ while cut 3 lies in the second BZ. The γ band is clearly observed in the cut 2 measurement but the α/β bands are weak (Fig. 3c). On the other hand, in the cut 3 measurement (Fig. 3d), the α/β bands are clearly observed but the γ band is weak. The observed band structures are different for the cut 1 and cut 4 because these two momentum cuts are not equivalent in the unfolded BZ.

Following the reviewer's suggestion, in the revised manuscript, we added polarization-dependent ARPES measurements to clarify the orbital character of the bands in $\text{La}_3\text{Ni}_2\text{O}_7$ and added a new figure (Supplementary Fig. 1) in Supplementary Information and replotted as Fig. R8 below. Based on the matrix element analysis, our results are consistent with the orbital assignment that the α/β bands are dominated by the dx^2-y^2 orbital while the γ band is dominated by the dz^2 orbital.

Fig. R8. Polarization-dependent ARPES measurements of $\text{La}_3\text{Ni}_2\text{O}_7$. **a** Fermi surface mapping measured under the s polarization geometry. The direction of the corresponding electric field vector E is marked by a double arrow near the bottom-left corner. **b** The corresponding constant energy contour at the binding energy of 30 meV. **c** The corresponding band structure measured along the $\bar{\Gamma} - \bar{X}$ direction. The location of the momentum cut is shown by the red line in **a**. **d-f** Same as **a-c** but measured under the p polarization geometry. The direction of the corresponding electric field vector E

consists of both in-plane component and out-of-plane component. In the upper-left inset of **b**, the schematic dx^2-y^2 orbital is shown.

Response to Reviewer #2's comments

In the manuscript, the authors performed an ARPES measurement on $\text{La}_3\text{Ni}_2\text{O}_7$ which had been found to show superconductivity near 80K under pressure. Compared with the first-principles calculations, the author found that the Ni-3d_{z²} orbital forms a flat band-like structure and is the most strongly correlated, which may be a key to understanding the high temperature superconductivity of this compound.

Superconductivity near 80K has a great impact on the field of condensed matter physics, and clarifying the electronic structure of $\text{La}_3\text{Ni}_2\text{O}_7$ is an essential and urgent task. Since ARPES is one of the most powerful tools for observing the electronic structure in a compound, the manuscript may contain valuable information for a wide range of audiences. However, the relation between the observed spectrum and the high temperature superconductivity is not clearly analyzed. In my view, the authors need to address the following points.

We thank the reviewer for the careful reviewing of our paper and providing constructive comments and suggestions to improve our paper. We thank the reviewer for pointing out the importance and significance of our work by stating “clarifying the electronic structure of $\text{La}_3\text{Ni}_2\text{O}_7$ is an essential and urgent task” and “Since ARPES is one of the most powerful tools for observing the electronic structure in a compound, the manuscript may contain valuable information for a wide range of audiences”.

1. All experiments in this manuscript were performed under ambient pressure, while the superconductivity was observed under high pressure. The authors just briefly address the effect of pressure in the discussion section, citing their previous publication. Suppose the authors want to stress the importance of the orbital-dependent correlation effect on superconductivity. In that case, they should expand their discussions further, such as how mass renormalization will change (or un-change) under pressure, etc.

For a new unconventional superconductor like $\text{La}_3\text{Ni}_2\text{O}_7$, the determination of its electronic structures is a prerequisite to establish theories to understand superconductivity. Understanding the electronic structure of $\text{La}_3\text{Ni}_2\text{O}_7$ at ambient pressure is important for understanding its electronic structure under high pressure. Since $\text{La}_3\text{Ni}_2\text{O}_7$ is a strongly correlated system, it is uncertain how well its electronic structures can be represented by the band structure calculations. Our first ARPES measurements are necessary and significant to provide direct electronic structure

information to check on the reliability of band structure calculations and establish theories to understand superconductivity in $\text{La}_3\text{Ni}_2\text{O}_7$.

Since it is difficult to measure the electronic structure of $\text{La}_3\text{Ni}_2\text{O}_7$ under high pressure by ARPES due to technical limitation, we carried out DFT calculation at 29.5 GPa and ambient pressure (as shown in Supplementary Fig. 2 and replotted as Fig. R9) to explain the change of electronic structure of $\text{La}_3\text{Ni}_2\text{O}_7$ with pressure. Band structure calculations indicate that the γ band derived from the Ni- $3d_{z^2}$ orbital undergoes a pronounced change in the energy position under pressure; it is below the Fermi level at ambient pressure but crosses the Fermi level under high pressure. On the other hand, its band width only slightly increases under high pressure primarily due to enhanced inter-layer interaction. The band renormalization effect is expected to exhibit a small change with pressure and the electron correlation effect remains strong under high pressure.

Fig. R9. **a** Calculated band structures of $\text{La}_3\text{Ni}_2\text{O}_7$ with $U=3.5$ eV at ambient pressure. **b** Calculated band structures of $\text{La}_3\text{Ni}_2\text{O}_7$ with $U=3.5$ eV at 29.5 GPa.

Following the reviewer’s suggestion, in the revised manuscript, we added more discussions on the band renormalization under high pressure: “Since it is difficult to measure the electronic structure of $\text{La}_3\text{Ni}_2\text{O}_7$ under high pressure by ARPES due to technical limitation, we carried out DFT+U calculation at 29.5GPa and compared it with that at ambient pressure (as shown in Supplementary Fig. 2). Band structure calculations indicate that the γ bands derived from the Ni- $3d_{z^2}$ orbital undergo a pronounced change in the energy position under pressure; it is below the Fermi level at ambient pressure but crosses the Fermi level under high pressure. On the other hand, its band width only slightly increases under high pressure primarily due to the enhanced inter-layer interaction. The band renormalization effect is expected to exhibit a small change with pressure and the electron correlation effect remains strong under high pressure. This is consistent with many theoretical proposals that the Ni- $3d_{z^2}$ orbital-derived flat band is expected to play the dominant role in generating superconductivity in $\text{La}_3\text{Ni}_2\text{O}_7$ [10, 16, 18, 19, 22, 24].” and “For a new unconventional superconductor

like $\text{La}_3\text{Ni}_2\text{O}_7$, the determination of its electronic structures is a prerequisite to establish theories to understand superconductivity. Understanding the electronic structure of $\text{La}_3\text{Ni}_2\text{O}_7$ at ambient pressure is essential for understanding its electronic structure under high pressure. Since $\text{La}_3\text{Ni}_2\text{O}_7$ is a strongly correlated system, it is uncertain how well its electronic structures can be represented by the band structure calculations. Our first ARPES measurements are necessary and significant to provide direct electronic structure information to check on the reliability of band structure calculations and establish theories to understand superconductivity in $\text{La}_3\text{Ni}_2\text{O}_7$.”

2. Computational details of how the authors treated on-site Coulomb interaction ($U=3.5\text{eV}$) on top of DFT calculation must be addressed.

Following the reviewer’s suggestion, we have added “the details of DFT+U method” in Method part in the revised manuscript “To address the effect of strong Coulomb interaction, we employed DFT+U method as described in Ref [54]. To determine the U parameter, we tested the U values with 3, 3.5 and 4 eV. The position of the γ band closely matches the experimental measurement (~ 50 meV below the Fermi level) when U is set to 3.5 eV. Consequently, an effective Hubbard $U = 3.5$ eV is taken for the 3d electrons of Ni cations in this work.”

3. The authors stress the orbital-selective Mott and Hund physics in the discussion section. Regarding this, is there any indication of Mott/Hund physics from the current experimental result?

Our ARPES measurements provide two important results. The first is the strong electron correlation of the Ni $3d_{z^2}$ orbitals derived γ band and the second is orbital-selective band renormalizations. Since Ni- $3d_{z^2}$ orbitals are half-filled in $\text{La}_3\text{Ni}_2\text{O}_7$, like the Cu- $3d_{x^2-y^2}$ orbitals in the parent cuprate compounds, the observed strong band renormalization of the γ band may be related to the Mott physics. On the other hand, DMFT calculations (Ref [30]) indicate that increasing the onsite Coulomb repulsion U alone does not significantly enhance band renormalization. It does not produce an obvious orbital-selectivity either. Only when the Hund’s rule coupling J is considered can the orbital-selective strong band renormalization be realized. Our observation of strong orbital-selective band renormalizations indicates that the Hund physics is at play in $\text{La}_3\text{Ni}_2\text{O}_7$.

Following the reviewer’s suggestion, in the revised manuscript, we added related discussion “Since Ni- $3d_{z^2}$ orbitals are half-filled in $\text{La}_3\text{Ni}_2\text{O}_7$, like the Cu- $3d_{x^2-y^2}$ orbitals in the parent cuprate compounds, the observed strong band renormalization of the γ band may be related to the Mott physics. On the other hand, DMFT calculations [30] indicate that increasing the onsite Coulomb repulsion U alone does not significantly enhance band renormalization. It does not produce an obvious orbital-selectivity either. Only when the Hund’s rule coupling J is considered can the orbital-

selective strong band renormalization be realized [30]. Our observation of strong orbital-selective band renormalizations indicates that the Hund physics is at play in $\text{La}_3\text{Ni}_2\text{O}_7$.”

Response to Reviewer #3's comments

This work reports an ARPES study on the recently discovered superconductor $\text{La}_3\text{Ni}_2\text{O}_7$, which shows a T_c as high as 80 K under high pressure. They successfully observed energy bands originating from the Ni $3d_{x^2-y^2}$ and Ni $3d_{z^2}$ orbitals and concluded that the d_{z^2} bands show stronger correlation than the $d_{x^2-y^2}$ bands. As this is the first ARPES study on the compound, which has attracted strong interest recently, and the experiment has been properly carried out, I recommend its publication in Nature Communication if the following points are reconsidered in a revised manuscript:

We thank the reviewer for the careful reviewing of our paper and providing constructive comments and suggestions to improve our paper. The reviewer nicely captured the main results of our work and its significance and impact by stating that “As this is the first ARPES study on the compound, which has attracted strong interest recently”. We also thank the reviewer for recommending its publication after we consider his/her comments and suggestions.

1. The authors state that they have included the effect of electron correlation in their LDA+U calculation. However, by the LDA+U method, one can include the effect of Coulomb interaction only on the mean-field level, and no electron correlation (i.e., fluctuations around the mean-field) can be included. Otherwise, the authors could not have used the LDA+U band structure as a starting point for estimating the band renormalization. Also, “the LDA/DFT+U method” has to be mentioned in the manuscript.

We thank the reviewer for the comment. We agree with the reviewer that by the LDA+U method, one can include the effect of Coulomb interaction only on the mean-field level, and no electron correlation can be included. Following the reviewer's comment, in the revised manuscript, we modified “The addition of sizable U in the band structure calculations indicates that there is a strong electron correlation in $\text{La}_3\text{Ni}_2\text{O}_7$ ” into “The addition of sizable U in the band structure calculations indicates that there is a strong **effect of Coulomb interaction** in $\text{La}_3\text{Ni}_2\text{O}_7$ ”. Also following the reviewer's suggestion, we mentioned the LDA+U methods in the revised manuscript by adding “**We carried out band structure calculations of $\text{La}_3\text{Ni}_2\text{O}_7$ based on the DFT+U method (see **Methods**)**”.

2. It has been known that, if the energy band is fully occupied, there is no correlation effects and hence no band renormalization. Therefore, because the d_{z^2} band is gapped and the lower band is fully occupied, estimating the band renormalization using the same method as that for a metallic band may need some remarks.

We thank the reviewer for this comment. It is usually considered that, if an energy band is fully occupied, there is no correlation effects and hence no band renormalization. It is therefore interesting that in $\text{La}_3\text{Ni}_2\text{O}_7$, although the $3d_{z^2}$ -derived flat band lies below the Fermi level, it exhibits strong band renormalization effect. Our density functional theory calculations reveal that, in this bilayer phase, the two nearest intra-layer Ni cations exhibit significant interlayer coupling through two $3d_{z^2}$ orbitals via the apical oxygen. This coupling arises from the quantum confinement of the NiO_2 bilayer within the structure, resulting in an energy splitting of Ni cations that makes the $3d_{z^2}$ bonding bands lower in energy and fully occupied. However, for each Ni cation, only one electron is occupied in the Ni $3d_{z^2}$ orbital. Consequently, the Ni $3d_{z^2}$ orbital is half-filled, leading to the effects of band renormalization in $\text{La}_3\text{Ni}_2\text{O}_7$.

Following the reviewer's suggestion, we added the above discussion in the revised manuscript: “It is usually considered that, if an energy band is fully occupied, there is no correlation effects and hence no band renormalization. It is therefore interesting that in $\text{La}_3\text{Ni}_2\text{O}_7$, although the $3d_{z^2}$ -derived flat band lies below the Fermi level, it exhibits strong band renormalization effect. Our density functional theory calculations reveal that, in this bilayer phase, the two nearest intra-layer Ni cations exhibit significant interlayer coupling through two $3d_{z^2}$ orbitals via the apical oxygen. This coupling arises from the quantum confinement of the NiO_2 bilayer within the structure, resulting in an energy splitting of Ni cations that makes the $3d_{z^2}$ bonding bands lower in energy and fully occupied. However, for each Ni cation, only one electron is occupied in the Ni $3d_{z^2}$ orbital. Consequently, the Ni $3d_{z^2}$ orbital is half-filled, leading to the effects of band renormalization in $\text{La}_3\text{Ni}_2\text{O}_7$.”

3. Any possible impurity phase that gave rise to the Fermi surface indicated in green? Since the ARPES data show that the lattice constant of the impurity phase is identical to $\text{La}_3\text{Ni}_2\text{O}_7$, an intergrowth of another Ruddlesden-Popper layers?

Following the reviewer's suggestion, we carried out more ARPES measurements. The results are added in the Supplementary Information as Supplementary Fig. 1 and replotted as Fig. R10. We carried out laser ARPES measurements on different places of the sample surface and also on different samples of $\text{La}_3\text{Ni}_2\text{O}_7$. Since our laser has a small spot size of $\sim 15\mu\text{m}$, it is possible to reduce the effect of impurity phase. As seen in Supplementary Fig. 1 and Fig. R10, we do not observe the extra feature seen in synchrotron ARPES measurement marked as dashed green line in Fig. 2a. These results further indicate that this extra feature is unlikely to be attributed to intrinsic electronic structures of $\text{La}_3\text{Ni}_2\text{O}_7$ but more likely to originate from impurity phase. Since the

ARPES data show that the lattice constant of the impurity phase is identical to $\text{La}_3\text{Ni}_2\text{O}_7$, it is possible that it is from an intergrowth of another Ruddlesden-Popper layers.

Fig. R10. High-resolution laser ARPES measurements of $\text{La}_3\text{Ni}_2\text{O}_7$. It is replotted from Supplementary Fig. 1 in the revised manuscript.

A minor comment:

4. It is true that the d_{z^2} character is dominant in the corner of the extended Brillouin zone scheme, but according to in the present folded zone scheme and the present experimental data, the zone "center" seems more realistic terminology?

We thank the reviewer for this comment. In order to compare with other materials like cuprate superconductors, and also because the folded bands are usually weaker than the original bands (Fig. R10), we prefer to use the extended Brillouin zone scheme. In this case, the γ band is dominant at the corner of the Brillouin zone.

Summary of Modifications

1. On page 3, line 71, we added “We carried out band structure calculations of $\text{La}_3\text{Ni}_2\text{O}_7$ based on the DFT+U method (see **Methods**)”.
2. On page 4, line 105, we added “and Supplementary Fig. 1” .
3. On page 4, line 122, we changed “electron correlation” into “effect of Coulomb interaction”.
4. On page 5, line 143, we added “We found that the energy position of the flat band slightly changes between 30~50 meV in different samples we measured. This is likely due to the sample inhomogeneity, particularly the oxygen content variation in $\text{La}_3\text{Ni}_2\text{O}_{7-\delta}$ [42].”
5. On page 5, line 150, we added “To investigate the orbital characters of different bands in $\text{La}_3\text{Ni}_2\text{O}_7$, we carried out polarization-dependent ARPES measurements as shown in Supplementary Fig. 1. Based on the analysis of the photoemission matrix element effects as described in Supplementary Note 1, the results are consistent with the orbital assignment that the α/β bands are dominated by the Ni-3dx²-y² orbital while the γ band is dominated by the Ni-3dz² orbital.”
6. On page 6, line 156, we added discussion about the temperature dependence of the γ flat band in $\text{La}_3\text{Ni}_2\text{O}_7$ as “In order to check on the nature of the γ flat band in $\text{La}_3\text{Ni}_2\text{O}_7$, we carried out temperature-dependent measurements of the band as shown in Fig. 4. These data are measured with 85eV photon energy. The temperature-dependent band structures along the $\bar{\Gamma}' - \bar{M} - \bar{\Gamma}'$ direction are shown in Fig. 4a and the extracted EDCs at $\bar{\Gamma}'(\pi, \pi)$ are plotted in Fig. 4c. The flat band changes little with temperature and stays at the similar position (~40meV) below the Fermi level over the temperature range of 18~163K. It was found that $\text{La}_3\text{Ni}_2\text{O}_7$ exhibits a resistivity anomaly at ~110K and a magnetic susceptibility anomaly at ~153K [40]. Our results indicate that the flat band in $\text{La}_3\text{Ni}_2\text{O}_7$ does not change in its energy position both across the resistivity anomaly temperature and across the susceptibility anomaly temperature. This is different from the case of $\text{La}_4\text{Ni}_3\text{O}_{10}$. In $\text{La}_4\text{Ni}_3\text{O}_{10}$, the flat band near (π, π) crosses the Fermi level above the resistivity anomaly temperature (~140K). But it opens a gap of ~20meV below the resistivity anomaly temperature and stays ~20meV below the Fermi level at low temperature [41]. The different behaviors of the flat band near (π, π) make $\text{La}_3\text{Ni}_2\text{O}_7$ distinct from $\text{La}_4\text{Ni}_3\text{O}_{10}$.”
7. On page 6, line 184, we added “It is usually considered that, if an energy band is fully occupied, there is no correlation effects and hence no band renormalization. It is therefore interesting that in $\text{La}_3\text{Ni}_2\text{O}_7$, although the Ni-3dz²-derived flat band lies below the Fermi level, it exhibits strong band renormalization effect. Our density functional theory calculations reveal that, in this bilayer phase, the two nearest intra-layer Ni cations exhibit significant interlayer coupling through two Ni-3dz² orbitals via the apical oxygen. This coupling arises from the quantum confinement of the NiO₂ bilayer within the structure, resulting in an energy splitting of Ni cations that makes the Ni-3dz² bonding bands lower in energy and

- fully occupied. However, for each Ni cation, only one electron is occupied in the Ni-3d_{z²} orbital. Consequently, the Ni-3d_{z²} orbital is half-filled, leading to the effects of band renormalization in La₃Ni₂O₇.”
8. On page 7, line 211, we added more discussion as “Since Ni-3d_{z²} orbitals are half-filled in La₃Ni₂O₇, like the Cu-3d_{x²-y²} orbitals in the parent cuprate compounds, the observed strong band renormalization of the γ band may be related to the Mott physics. On the other hand, DMFT calculations [30] indicate that increasing the onsite Coulomb repulsion U alone does not significantly enhance band renormalization. It does not produce an obvious orbital-selectivity either. Only when the Hund’s rule coupling J is considered can the orbital-selective strong band renormalization be realized [30]. Our observation of strong orbital-selective band renormalizations indicates that the Hund physics is at play in La₃Ni₂O₇.”
 9. On page 8, line 220, we added more discussion as “Since it is difficult to measure the electronic structure of La₃Ni₂O₇ under high pressure by ARPES due to technical limitation, we carried out DFT+ U calculation at 29.5GPa and compared it with that at ambient pressure (as shown in Supplementary Fig. 2). Band structure calculations indicate that the γ bands derived from the Ni-3d_{z²} orbital undergo a pronounced change in the energy position under pressure; it is below the Fermi level at ambient pressure but crosses the Fermi level under high pressure. On the other hand, its band width only slightly increases under high pressure primarily due to the enhanced inter-layer interaction. The band renormalization effect is expected to exhibit a small change with pressure and the electron correlation effect remains strong under high pressure. This is consistent with many theoretical proposals that the Ni-3d_{z²} orbital-derived flat band is expected to play the dominant role in generating superconductivity in La₃Ni₂O₇ [10, 16, 18, 19, 22, 24].” and “For a new unconventional superconductor like La₃Ni₂O₇, the determination of its electronic structures is a prerequisite to establish theories to understand superconductivity. Understanding the electronic structure of La₃Ni₂O₇ at ambient pressure is essential for understanding its electronic structure under high pressure. Since La₃Ni₂O₇ is a strongly correlated system, it is uncertain how well its electronic structures can be represented by the band structure calculations. Our first ARPES measurements are necessary and significant to provide direct electronic structure information to check on the reliability of band structure calculations and establish theories to understand superconductivity in La₃Ni₂O₇.”
 10. On page 8, line 250, in Method, we added “and BL03U”.
 11. On page 9, line 255, in Method, we added “The momentum coverage is increased by applying bias on the sample during the ARPES measurements [53] (Supplementary Fig. 1).”
 12. On page 9, line 271, in Method, we added “To address the effect of strong Coulomb interaction, we employed DFT+ U method as described in Ref [59]. To determine the U parameter, we tested the U values with 3, 3.5 and 4eV. The position of the γ band closely matches the experimental measurement (\sim 50meV

below the Fermi level) when U is set to 3.5 eV. Consequently, an effective Hubbard $U = 3.5\text{eV}$ is taken for the 3d electrons of Ni cations in this work.”

13. We added a new Figure 4 in revised manuscript to discuss the temperature dependence of the flat band in $\text{La}_3\text{Ni}_2\text{O}_7$.
14. Following the reviewer’s suggestion, we modified the Figure 5.
15. Following the reviewer’s suggestion, we added a new Supplementary Figure 1 and Supplementary note 1 in Supplementary Information to discuss the orbital character of bands in $\text{La}_3\text{Ni}_2\text{O}_7$.
16. Following the reviewer’s suggestion, we added a new Supplementary Figure 2 to discuss the evolution of flat band under pressure.

REVIEWERS' COMMENTS

Reviewer #1 (Remarks to the Author):

The authors made a great effort in this reply, and have addressed most of my concerns. It is particularly interesting to see any lack of temperature dependence in the spectra across the said resistive anomalies in the 327 system. However, I am not convinced that the temperature dependent data can allay the concern of hastily tying shallow flat band to superconductivity. After all, even with the gapped band as shallow as 20 meV below EF in the 4310 system (even shallower than here in 327), it is not really superconducting there. While the comparison of the temperature dependence adds another layer of intrigue to the work, the link between flat dz2 band and superconducting remains weak if not weaker. I would suggest the authors being less biased towards this theoretical speculation (admittedly a very popular speculation), and be more objective about the data interpretation.

The new laser-ARPES data is a welcome addition to add substance to the speculated impurity phase. However, caveats remain. It is not uncommon to have states that won't be visible at low photon energies (e.g. arXiv:2211.08114), and only becomes visible at certain photon energies due to the final state or initial state matrix element. In this regard, the new data is not able to convincingly demonstrate these being impurity states. Again, since these states follow almost the same BZ size and symmetry as the primary bands, I stand by my suspicion that they may come from surface reconstructions or inherent super-modulations. In this regard, the authors should clarify what they mean by "impurity" - is it a twin phase, a superstructure, a completely new chemical impurity phase, or surface reconstruction? Clarifying this aspect will be essential to achieving the primary mission and impact of this work (i.e. lay out a definitive experimental validation of the basic electronic structure of this new system).

I shall leave the final judgement to the other reviewers.

Reviewer #2 (Remarks to the Author):

The authors have responded to my previous comments and revised the manuscript satisfactorily, so I recommend its publication in Nature Communications.

Reviewer #3 (Remarks to the Author):

The authors have responded to my comments and revised the manuscript accordingly. Now I recommend the publication of the paper in Nature Communications.

Response to Reviewer's Comments

Response to Reviewer #1's comments

The authors made a great effort in this reply, and have addressed most of my concerns. It is particularly interesting to see any lack of temperature dependence in the spectra across the said resistive anomalies in the 327 system. However, I am not convinced that the temperature dependent data can allay the concern of hastily tying shallow flat band to superconductivity. After all, even with the gapped band as shallow as 20 meV below E_F in the 4310 system (even shallower than here in 327), it is not really superconducting there. While the comparison of the temperature dependence adds another layer of intrigue to the work, the link between flat d_{z^2} band and superconducting remains weak if not weaker. I would suggest the authors being less biased towards this theoretical speculation (admittedly a very popular speculation), and be more objective about the data interpretation.

We thank the reviewer for reviewing our revised manuscript again. In our paper, we mentioned that “in many theoretical proposals, this band (flat band) is expected to play the dominant role in generating superconductivity in $\text{La}_3\text{Ni}_2\text{O}_7$ ”. This is just an objective description of the present situation. The role of the flat band in generating superconductivity in $\text{La}_3\text{Ni}_2\text{O}_7$ definitely needs further experimental and theoretical efforts.

The new laser-ARPES data is a welcome addition to add substance to the speculated impurity phase. However, caveats remain. It is not uncommon to have states that won't be visible at low photon energies (e.g. arXiv:2211.08114), and only becomes visible at certain photon energies due to the final state or initial state matrix element. In this regard, the new data is not able to convincingly demonstrate these being impurity states. Again, since these states follow almost the same BZ size and symmetry as the primary bands, I stand by my suspicion that they may come from surface reconstructions or inherent super-modulations. In this regard, the authors should clarify what they mean by "impurity" - is it a twin phase, a superstructure, a completely new chemical impurity phase, or surface reconstruction? Clarifying this aspect will be essential to achieving the primary mission and impact of this work (i.e. lay out a definitive experimental validation of the basic electronic structure of this new system).

We carried out ARPES measurements on $\text{La}_3\text{Ni}_2\text{O}_7$ many times. The extra feature was observed only once when the spot size is relatively large but was not observed in other measurements when the spot size is small. These indicate that the extra feature is most likely from the impurity phase. Since the extra feature follows the same BZ as $\text{La}_3\text{Ni}_2\text{O}_7$, it is possible that it comes from intergrowth phase.

Following the reviewer's suggestion, we added in the revised manuscript "Since the extra feature follows the same Brillouin zone as $\text{La}_3\text{Ni}_2\text{O}_7$, it is possible that it comes from intergrowth phase."

I shall leave the final judgement to the other reviewers.

We thank the reviewer again for carefully reviewing our paper and providing constructive comments and suggestions to improve our paper.

Response to Reviewer #2's comments

The authors have responded to my previous comments and revised the manuscript satisfactorily, so I recommend its publication in Nature Communications.

We thank the reviewer for reviewing our revised manuscript and recommending its publication in Nature Communications.

Response to Reviewer #3's comments

The authors have responded to my comments and revised the manuscript accordingly. Now I recommend the publication of the paper in Nature Communications.

We thank the reviewer for reviewing our revised manuscript and recommending its publication in Nature Communications.